# Microbiological quality and genotoxicity of domestic water sources: A combined approach using Micro Biological Survey method and mutagenesis assay (micronucleus test) in root tips of *Vicia faba* in the West region of Cameroon

**Rodrigue Biguioh Mabvouna**[1]*, **Sali Ben Béchir Adogaye**[1], **Patrick Martial Pete Nkamedjie**[2], **Andrillene Laure Deutou Wondeu**[3], **Martin Sanou Sobze**[4], **Jean Blaise Kemogne**[3], **Vittorio Colizzi**[1]

**1** Faculty of Medicine and Surgery, University of Roma "Tor Vergata", Roma, Italy, **2** Institute for Research, Socio-Economic Development and Communication (IRESCO), Yaoundé, Cameroon, **3** Evangelic University of Cameroon, Mboua-Bandjoun, Cameroon, **4** Faculty of Medicine and Pharmaceutical Sciences, University of Dschang, Dschang, Cameroon

* rodrigue.biguioh@yahoo.fr

## Abstract

At least 2.1 billion people around the world use contaminated drinking water, causing 485,000 diarrheal deaths each year, mostly among children under 5 years old. A study conducted 10 years ago in Bafoussam (West Cameroon) recorded concentrations of bacteria among surface and groundwater. High levels of bicarbonates, phosphates, chlorides and suspended matters were also found. The aim of this study was to assess the microbiological and chemical qualities of domestic water sources in 5 localities of the West region of Cameroon. Water samples from 22 water sources (wells, springs, water drilling and river) were aseptically collected in plastic bottles and transferred in 50 ml sterile tubes. For chlorinated water sources, 1 ml of Thiosulfate was added to the water sample; immediately placed in an ice box and transported to the laboratory for analysis. Water temperature and pH were measured on site. The microbiological quality of water was determined by testing Total Coliforms (TC) using the Micro Biological Survey method. 1 ml of each water sample was inoculated in the MBS vial initially rehydrated with 10 ml of sterile distilled water. The initial color of the vials is red. Color changes were monitored at three different time intervals (12h, 19h and 24h), corresponding to three levels of contamination. The chemical quality of water was assessed using micronucleus (MN) test in selected *Vicia faba* seeds secondary root tips permanently mounted in Dibutylphthalate Polystyrene Xylene mountant for histology after 72 hours of direct exposition in water samples and in dark. The mitotic indices and MN frequencies were evaluated in 10 root tips per site analysing 5000 cells per tip. Statistical analyses were done using Stata IC/15.0 software. The Student *t*-test was used for mean comparison and the significance level was set at 1%. The majority of samples were collected from wells (63.6%). The mean water pH ranged from 5.5 to 8.3 and the temperature

**Data Availability Statement:** All relevant data are within the paper and supporting information files.

**Funding:** This study was supported by Bread for the World (Brot für die Welt). https://www.brot-fuer-die-welt.de/. The funders had no role in study design, data collection and analysis, decision to publish, or preparation of the manuscript.

**Competing interests:** The authors have declared that no competing interests exist.

**Abbreviations:** CFU, Colony Forming Units; HACCP, Hazard Analysis Critical Control Point; MBS, Micro Biological Survey; MN, micronucleus; TC, Total Coliforms; WHO, World Health Organization.

varied from 23 to 26˚C. A very high concentration of TC [$>10^3$ CFU/ml] was found in 8 (36.4%) samples. 10 (45.5%) and 2 (9.1%) samples turned yellow at 19 and 24 hours respectively after incubation corresponding to TC concentration of [$10<x<10^3$ CFU/ml] and [$1<x<10$ CFU/ml]. The MN frequency was higher ($P \leq 0.01$) compared to the negative control in 9 (40.9%) water samples indicating significant genotoxic effects of these water sources. This study highlighted the poor quality of domestic water sources in West region of Cameroon and the need to conduct regular monitoring of drinking water sources. Community capacity building on water treatment methods, including good wastes management should be implemented to help improve water quality.

## Introduction

Adequate, accessible and safe drinking water source is often regarded as an important means of disease prevention [1]. Improving access to safe drinking-water can result in tangible benefits to health [2]. According to the World Health Organization (WHO) guidelines for drinking-water quality, safe water should not represent any significant risk to health over a lifetime of consumption, including different sensitivities that may occur between life stages [2]. In 2017, the WHO estimated that 785 million people lacked a basic drinking water service, including 144 million people who were dependent on surface water [3]. Infants and young children are the most exposed to waterborne diseases, especially those living in unsanitary conditions. At least 2.1 billion people around the world use a drinking water source contaminated with faeces; and according to forecasts, half of the world's population will be living in water-stressed areas by 2025 [3].

Water contaminants can be defined as any physical, chemical, biological, or radiological substance or material in water [2, 3]. Contaminants may be harmful if consumed at certain levels in drinking water [4]. Microorganism and chemical contaminants can transmit diseases such diarrhoea, cholera, dysentery, typhoid, cancers, infertility, immunological and neurological diseases [2–5]. Contaminated drinking water is estimated to cause 485,000 diarrheal deaths each year, mostly among children under 5 years old [3, 6]. There are many sources of contamination of drinking water resulting from domestic, agricultural or industrial activities [2–7]. Previously, attention was mainly focused on the microbiological quality of drinking water, but with the development of toxicology and the increase of knowledge on the risks of chemicals, studies of chemical quality of drinking water have increased. The microbiological quality of drinking water can be assessed by monitoring faecal indicator bacteria, such as Coliform bacteria [8–10].

Coliforms are a group of bacteria that are generally not harmful for human beings. They are present in the digestive tracts of humans and animals, thus in their faeces. Generally, the concentrations of faecal bacteria in water sample are low, and the quantity of possible bacteria present in the sample is high. Thus for water quality analyses, it is not practical to test the presence of each pathogen in the sample. As a result, the presence of other harmful biological organisms can be indirectly detected by testing Total Coliforms [11, 12]. Total Coliform bacteria are used as an indicator of other pathogenic bacteria mainly because their presence in water sample indicates environmental contamination and presence of water borne diseases bacteria [13]. Coliform bacteria also allow to assess the efficiency of water treatment (disinfection, chlorination or boiling), thus their presence indicates an insufficient, inadequate or absence of water treatment [14]. In addition to microbiological quality, chemical substances from human activities can also affect the quality of drinking water sources [15, 16].

In limited resources settings, the key issue in water safety policy is how to determine the microbiological and chemical qualities of drinking water with simple but valid methods. From a public health perspective, simple routine checking of drinking water should be designed to detect the presence or absence of microorganisms in water and the level of toxicity that can affect human health. In Cameroon, there are few studies assessing both microbiological and chemical qualities of drinking water sources in rural or urban settings. A study conducted 10 years ago in Bafoussam (West region) recorded higher concentrations of bacteria such as *Escherichia coli*, *Salmonella*, *Shigella* among surface and groundwater. Chemically, high levels of bicarbonates, phosphates, chlorides, and suspended materials were found in rivers and springs water samples [17]. To provide updated data and evidences for steering public health decisions, we performed a combined approach to assess the microbiological quality of domestic water sources by the detection of Total Coliforms (TC) using Micro Biological Survey (MBS) method and the chemical quality was assessed using the mutagenesis assay by the detection of genotoxic effects in root tips of *Vicia faba*.

## Materials and methods

### Study location

The study was conducted in the West region of Cameroon which has an estimated population of 1,921,590 inhabitants (from the 2008 population Census) [18]. Inhabitants are mostly farmers (coffee, potatoes, maize, beans, and vegetables production) and their farming activities require the use of fertilizers that could potentially pollute the environment including water sources.

### Study design, period and water sampling

This was a cross-sectional study carried-out in five localities of west Cameroon (Bafoussam 1er, Bafoussam 2e, Foumbot, Galim and Kouoptamo) between October and November 2018. 22 samples from domestic water sources (wells, spring, drilling water and river) used as drinking water by local populations were aseptically collected in the morning before 12 o'clock. At each sampling site, 1 litre of water was directly collected in sterile plastic bottles. 50 ml of sample was taken in plastic tubes from collected water and 1 ml of Sodium Thiosulfate solution 10% (0.25g/50ml) was added to the sample; for chlorinated water sources. The tube containing the sample was immediately placed in an ice box and then transported to the laboratory for microbiological and genotoxic analysis. Temperature and pH of water were measured on site before placing the samples in the ice box. Water sources to be sampled were identified step by step according to their availability, and accessibility. Table 1 presents details on the water sources collected for analyses.

### Total Coliforms detection

Quantitative detection of Coliforms in water samples was done using MBS-HACCP & water Easy test. This test is simple and can be performed by anyone, anywhere in an error-free way. The MBS method is a colorimetric reaction system based on the measurement of the catalytic activity of oxidoreductase enzymes of primary metabolism, which allows to determine a correspondence between microorganism and enzyme activity in the sample [19]. This method has been applied in similar context in Africa including Cameroon for the assessment of the microbiological safety of drinking water [20, 21], and had been found to be mostly adapted for developing countries as it requires less skills, time and can be done at low cost. The test comes with all the material necessary and the analysis doesn't require other reagents or instruments. The MBS method is based on the visual observation of the color change in the suspension formed in the analysis vial used when the test sample is inoculated [19]. The color change (from red to

**Table 1. Sampling locations, water sources categories, sample temperature (˚C), geographic coordinates, and samples collection dates.**

| Sampling location | Water source category | Sample temperature (˚C) | Geographic coordinates | | Collection date |
|---|---|---|---|---|---|
| | | | Latitude | Longitude | |
| Foumbot | Protected spring | 24 | 5,5783333 | 10,6391667 | 23/09/2018 |
| | River | 24 | 5,5866667 | 10,6502778 | 23/09/2018 |
| | Protected well | 23 | 5,5558333 | 10,6125 | 24/09/2018 |
| | Unprotected well | 23 | 5,5722222 | 10,6158333 | 24/09/2018 |
| | Unprotected well | 23 | 5,4947222 | 10,6047222 | 24/09/2018 |
| | Protected spring | 24 | 5,4969444 | 10,5991667 | 24/09/2018 |
| | Protected well | 24 | 5,5422222 | 10,5930556 | 24/09/2018 |
| Bafoussam 1er | Unprotected spring | 24 | 5,4866667 | 10,5180556 | 23/09/2018 |
| | Unprotected well | 23 | 5,5116667 | 10,4933333 | 25/09/2018 |
| | Drilling water | 23 | 5,5008333 | 10,5216667 | 28/09/2018 |
| Bafoussam 2e | Protected spring | 24 | 5,5636111 | 10,5386111 | 24/09/2018 |
| | Protected well | 23 | 5,5530556 | 10,5225 | 28/09/2018 |
| Galim | Protected well | 23 | 5,7280556 | 10,5225 | 23/09/2018 |
| Kouoptamo | Unprotected well | 26 | 5,6211111 | 10,4783333 | 23/09/2018 |
| | Unprotected well | 23 | 5,6186111 | 10,6252778 | 25/09/2018 |
| | Unprotected well | 22 | 5,645 | 10,5663889 | 25/09/2018 |
| | Protected well | 22 | 5,6344444 | 10,5447222 | 25/09/2018 |
| | Protected spring | 23 | 5,7394444 | 10,5788889 | 28/09/2018 |
| | Protected well | 23 | 5,7394444 | 10,6097222 | 28/09/2018 |
| | Protected spring | 23 | 5,6825 | 10,595 | 28/09/2018 |
| | Unprotected well | 23 | 5,6816667 | 10,5941667 | 28/09/2018 |
| | Unprotected well | 23 | 5,6486111 | 10,5611111 | 28/09/2018 |

yellow) occurs when the water sample added in the vial contains Coliforms bacteria. The changes in color according to time are proportional to the concentration of coliforms in the water samples, the greater the amount of microorganisms, the more rapid the change of color, thus, a positive result (contamination) [19–21]. The concentration of bacteria is expressed in Colony Forming Units per millilitres of water (CFU/ml) for the analysis water samples.

Before starting with the analysis, the Coliforms MBS (Coli MBS) vials were rehydrated with 10 ml of sterile distilled water and shaken to dissolve the reagent. 1 ml of each water sample was collected from plastic tubes using a sterile Pasteur pipette and inoculated in the Coli MBS vial. The vials were then closed and shaken for about 30 seconds for homogenization. Each analysis was performed twice and the vials were incubated at 37˚C. The color changes of the Coli MBS reaction vials were monitored using the chromatic scale according to the standard protocol at three different time intervals (12 hours, 19 hours, and 24 hours), corresponding to three levels of contamination [19]. The initial color of the Coli MBS vials is red. A color change from red to yellow after 12 hours indicates a very high contamination (TC concentration $> 10^3$ CFU/ml); a color change at 19 hours indicates a high contamination (TC concentration $10 < x < 10^3$ CFU/ml) and a color change at 24 hours corresponds to TC concentration of $1 < x < 10$ CFU/ml. For the negative control, 1 ml of distilled water was inoculated in the Coli MBS vial [19].

## Micronucleus (MN) test

The MN test is considered to be one of the most suitable methods for identifying response to exposure to a complex mixture of contaminants [22, 23]. This test is used both as a short-term

test in animals, humans and for the detection of the mutagenic potential of pollutants in air, water and soil, in environmental monitoring programs [24]. Its application in plant systems (*Vicia faba*, *Allium cepa*, *Tradescantia sp*. etc.) is particularly indicated for the detection of mutagens present in water and soil [25–28], allowing the analysis of raw environmental matrices without purification or concentration processes [22, 23]. For its sensitivity and reliability, MN test in *Vicia faba* roots is one of the most widely used methods in aquatic genotoxicity studies, also limiting the use of animal testing [29]. This tests is able to detect both clastogenic and aneugenic effects. Finally, its low cost and its simplicity of execution make it particularly suitable for monitoring polluted fresh water.

## Mutagenesis assay in *Vicia faba* root tips

The MN test was performed in the secondary root tips of *Vicia faba* according to standard protocol [29, 30]. Micronuclei occur as small nuclei next to the main nucleus of interphase cells and their appearance is related to the formation of chromosome fragments or the loss of whole chromosomes during cell division (mitotic). They arise both from the loss of chromosome fragments, due to the absence of the centromere, and from the loss of entire chromosomes, due to induced functional damage to the mitotic spindle or the inhibition of the functions of other structures, including the interaction between the centromere-kinetochore and the fibers of the mitotic spindle involved in the process of segregation. Once induced, the lost fragment and chromosome, not binding to the fibers of the spindle, are excluded from the process of segregation.

## Mutagenesis assay operating procedures

About 400 dry *Vicia faba* seeds were rehydrated for 24 hours in tap water. The seeds coats were removed and the seeds placed in moist cotton for germination at ambient temperature in the dark. Four to five days after seeding, the primary roots, about 3 cm in length were selected and their tips were cut off. For each sample, five seeds were selected. To perform the test, the primary roots of selected seeds were suspended in water samples contained in glass containers at ambient temperature and in the dark. Distilled water was used as negative control. No positive control was used for the experiment. After three days (72 hours), secondary roots of about 1–2 cm were obtained, cut off and fixed with a 3:1 solution of ethyl alcohol/acetic acid. After 30 minutes, the fixing solution was replaced by a new one and the roots were stored at + 4°C after covering each container with parafilm. Before staining, the roots were transferred from the fixing solution to tap water for 10 to 15 minutes at ambient temperature. The rehydrated roots were transferred to a preheated (60°C) 1N chloridic acid solution and placed in a water bath for 10 minutes. After complete removal of the chloridic acid solution, the roots were immersed in the Schiff reagent for 45 to 60 minutes at ambient temperature and in the dark. Stained roots were squashed on to slides in 45% acetic acid, then immerse slowly in liquid nitrogen and the coverslips were quickly removed using a scalpel blade. The slides were air-dried for 8 to 10 hours or overnight and permanently mounted in DPX mountant for histology. The mitotic indices and MN frequencies were evaluated in 10 root tips per site analysing 5000 cells per tip. The Student *t*-test was used to perform comparison between the average MN frequency at each site and the negative control. Statistical analyses were done using Stata IC/ 15.0 software (College Station, Texas 77845 USA, http://www.stata.com) and the significance level was set at 1% for the MN frequency and at 5% for proportions.

## Ethical approval

This study does not report on or involve the use of any animal or human data or tissue, so ethical approval and consent were not required with reference to Order No. 079/A/MSP/DS of the

Minister of Public Health of October 22, 1987 establishing and organizing an Ethics Committee on Research Involving Human Beings (article 2).

## Results

The majority of the samples were collected from wells water (63.6%) and springs water (27.2%). The pH of the water samples collected ranged from 5.5 to 8.3 and the temperature varied from 23 to 26˚C.

### Microbiological quality

Concerning the microbiological quality of the samples, all the four types of water sources were contaminated with TC, except two wells where samples were found to be safe [TC counts <1 CFU/100 ml], (Table 2). A very high concentration of TC [$>10^3$ CFU/ml] was found in 8 (36.4%) samples. 10 (45.5%) and 2 (9.1%) water samples turned yellow at 19 hours and 24 hours after incubation corresponding to TC concentration of [$10<x<10^3$ CFU/ml] and [$1<x<10$CFU/ml] respectively. The highest TC count was observed independently of the types of water sources analysed. Water sources form the sampling site of Kouoptamo appeared more contaminated. All the 9 samples of this locality were contaminated, 2 of them with very high TC concentration [$>10^3$ CFU/ml].

### Chemical quality

Regarding the chemical quality of the water samples, the mutagenesis assay in root tips of *Vicia faba* revealed a significant genotoxicity in almost half of the samples (40.9%), Fig 1.

**Table 2. Total Coliforms concentration in water samples, level of contamination and time of analysis, West region, Cameroon, 2018.**

| Sampling location | Water source category | Total Coliform concentration (CFU/mL) | Level of contamination | Duration of analysis (hours) |
|---|---|---|---|---|
| Foumbot | Protected spring | $1 < x < 10$ | Low | 24 |
| | River | $> 10^3$ | Very high | 12 |
| | Protected well | <1 | Very low | No color change |
| | Unprotected well | $10 < x < 10^3$ | High | 19 |
| | Unprotected well | $> 10^3$ | Very high | 12 |
| | Protected spring | $1 < x < 10$ | Low | 24 |
| | Protected well | $10 < x < 10^3$ | High | 19 |
| Bafoussam 1er | Unprotected spring | $> 10^3$ | Very high | 12 |
| | Unprotected well | $10 < x < 10^3$ | High | 19 |
| | Drilling water | $10 < x < 10^3$ | High | 19 |
| Bafoussam 2e | Protected spring | $> 10^3$ | Very high | 12 |
| | Protected well | <1 | Very low | No color change |
| Galim | Protected well | $> 10^3$ | Very high | 12 |
| Kouoptamo | Unprotected well | $> 10^3$ | Very high | 12 |
| | Unprotected well | $10 < x < 10^3$ | High | 19 |
| | Unprotected well | $10 < x < 10^3$ | High | 19 |
| | Protected well | $> 10^3$ | Very high | 12 |
| | Protected spring | $10 < x < 10^3$ | High | 19 |
| | Protected well | $10 < x < 10^3$ | High | 19 |
| | Protected spring | $10 < x < 10^3$ | High | 19 |
| | Unprotected well | $10 < x < 10^3$ | High | 19 |
| | Unprotected well | $10 < x < 10^3$ | High | 19 |

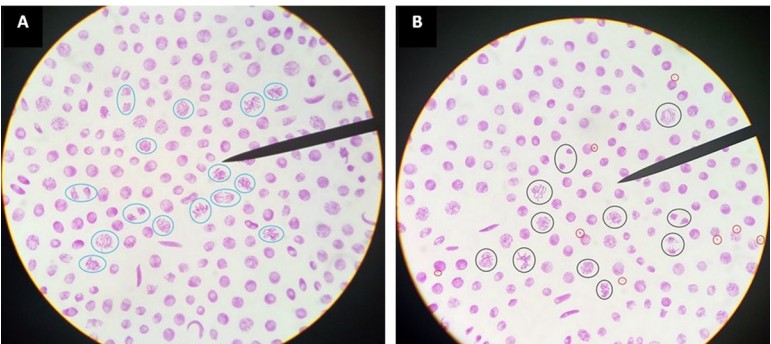

**Fig 1.** *Vicia faba* cells in proliferation: (A) non genotoxic water source slide, mitotic cells are circled in blue; (B) genotoxic water source slide, mitotic cells are circled in black and micronuclei are circled in red.

Globally, the mean cells proliferation including mitotic cells of the examined root tips was similar to the negative control and the mitotic index values ranged from 5.4% and 8.8% (Table 3). The MN frequency was significantly higher ($P \leq 0.01$) than the negative control value in 35.7% of wells water, 50% of springs and in the only river analysed. The Student *t*-test showed significant genotoxic effects in 9 water samples, of which 5 (55.6%) wells, 3 (33.3%) springs and a river (11.1%), (Table 4).

**Table 3. Mean per 5000 cells ± Std. Err. of mitotic cells and mitotic index in root tips of *Vicia faba* exposed to water samples, West region, Cameroon, 2018.**

| Sampling location | Sources | Mitotic cells | | Mitotic index (%) | |
|---|---|---|---|---|---|
| | | Mean | SE | Mean | SE |
| Foumbot | Spring | 365 | ±16.98 | 7.3 | ±0.3 |
| | River | 349 | ± 12.3 | 6.9 | ±0.2 |
| | Well | 277 | ±13.7 | 5.5 | ±0.2 |
| | Well | 278 | ±51.2 | 5.5 | ±1 |
| | Well | 316 | ±52.4 | 6.3 | ±1 |
| | Spring | 293 | ±25.8 | 5.8 | ±0.5 |
| | Well | 273 | ±25.1 | 5.4 | ±0.5 |
| Bafoussam 1er | Spring | 321 | ±12.38 | 6.4 | ±0.2 |
| | Well | 339 | ±41.5 | 6.7 | ±0.8 |
| | Water drilling | 385 | ±43.8 | 7.7 | ±0.8 |
| Bafoussam 2e | Spring | 284 | ±38.9 | 5.6 | ±0.7 |
| | Well | 374 | ±41.1 | 7.4 | ±0.8 |
| Galim | Well | 293 | ±18.9 | 5.8 | ±0.3 |
| Kouoptamo | Well | 314 | ±22.7 | 6.2 | ±0.4 |
| | Well | 317 | ±17.1 | 6.3 | ±0.3 |
| | Well | 348 | ±24.9 | 6.9 | ±0.4 |
| | Well | 343 | ±35.1 | 6.8 | ±0.7 |
| | Spring | 367 | ±22.7 | 7.3 | ±0.4 |
| | Well | 279 | ± 23.3 | 5.5 | ±0.4 |
| | Spring | 441 | ± 39.3 | 8.8 | ±0.7 |
| | Well | 378 | ±45.8 | 7.5 | ±0.9 |
| | Well | 349 | ±45.9 | 6.9 | ±0.9 |
| **Total** | - | **331** | **±9.3** | **6.6** | **±0.1** |
| **Negative control** | **Tap water** | **332** | **±16.5** | **6.6** | **±0.3** |

**Table 4. Mean per 5000 cells ± Std. Err. of MN frequency in root tips of *Vicia faba* exposed to water samples, West region, Cameroon, 2018.**

| Sampling location | Sources | Mean | SE | [99% CI] |
|---|---|---|---|---|
| Foumbot | Spring | 4.1 | ±0.52 | [2.3–5.8] |
| | River | 11.8* | ±0.6 | [10.2–13.3] |
| | Well | 3.9 | ±0.3 | [2.6–5.1] |
| | Well | 4.3 | ±0.4 | [2.7–5.8] |
| | Well | 3.6 | ±0.4 | [2.2–4.9] |
| | Spring | 4.1 | ±0.2 | [3.2–4.9] |
| | Well | 3.6 | ±0.3 | [2.6–4.5] |
| Bafoussam 1er | Spring | 13.6* | ±0.8 | [11.7–16.4] |
| | Well | 4.1 | ±0.5 | [2.4–5.7] |
| | Water drilling | 4 | ±0.4 | [2.5–5.4] |
| Bafoussam 2e | Spring | 13.8* | ±0.8 | [11.1–16.4] |
| | Well | 4.2 | ±0.3 | [2.9–5.4] |
| Galim | Well | 4 | ±0.3 | [2.7–5.2] |
| Kouoptamo | Well | 4 | ±0.2 | [3.1–4.8] |
| | Well | 3.8 | ±0.2 | [2.8–4.7] |
| | Well | 10.5* | ± 1.4 | [5.8–15.1] |
| | Well | 9.6* | ±0.7 | [7–12.1] |
| | Spring | 10.8* | ±1.2 | [6.6–14.9] |
| | Well | 9.7* | ±0.9 | [6.6–12.7] |
| | Spring | 4 | ±0.4 | [2.4–5.5] |
| | Well | 12.1* | ±1.3 | [7.5–16.6] |
| | Well | 8.9* | ±1.2 | [4.9–12.8] |
| **Negative control** | **Tap water** | **3.4** | **±0.26** | **[2.7–4]** |

*The MN frequency is significantly higher (P ≤ 0.01) than the negative control value.

## Discussion

Water from wells, drilling, spring and river are the major domestic water sources in sub-Saharan Africa and therefore are essential resource for human activities and for consumption. The control of their quality remains a concern in many countries. Water represents one of the principal sources of diseases transmission and the use of unsafe water can cause serious health problems due to the presence of potential contaminants. The use of simple and rapid methods for both microbiological and chemical water qualities check that can be performed anywhere even by unskilled personnel is one of the key components of effective water management policy, especially in resource-limited countries. In this study, we used a combined approach for water microbiological and chemical qualities check. MBS method results showed that almost all types of water sources contained TC and higher concentrations of TC was found among wells. Our findings are consistent with the results of previous study in Bafoussam (West Cameroon), which reported high concentration of TC in groundwater (well water and spring water) mostly during rainy season with TC concentration higher than $10^3$ CFU/ml [17]. Our study was also conducted during rainy season suggesting possible high bacteria growth in groundwater than in surface water (running water) in this period. This could be favoured by the combined effect of high temperatures (from 23 to 26°C) and pH (from 5.5 to 8.3). It is also important to note that most of wells were closed to latrines and/or human habitation and hence phreatic slicks of wells could communicate with latrines during rainy season due to additional water flow. External contamination by runoff water might also explain high TC

concentration observed as bacteria could be transported by water from latrines and/or dumps to wells. Additional investigations are required to further assess these relations. In 20 (90.9%) water samples, TC counts were above the recommended levels (<1 CFU/100ml) set by WHO [2, 31], including treated sources (chlorinated sources) indicating an insufficient or inadequate treatment of the sources. However, the TC concentration of untreated source water was found to be higher than treated water sources with no statistically significant difference. Coliform bacteria and other harmful biological organisms have the same origin. The detection of coliforms is simple and rapid. Their quantity in water sample is higher than other pathogenic bacteria. As other dangerous organisms, coliforms are sensitive to water treatment such as disinfection, boiling, thus assessing coliforms as indicator of microbiological quality of drinking water can be a reasonable approach of testing the presence or absence of other pathogenic bacteria in water.

We also applied MN test in *Vicia faba* roots tips as a first alarm system for mutagenic effects of domestic waters sources. The genotoxicity was measured by direct exposition of *Vicia faba* roots to water samples. Other methods using plants such as *Tradescantia* [32–34]; aquatic animals e.g. amphibian, fishes, mussels [24, 35, 36] exist and can be used with the difference that they are more complex to perform compared to the MN test in *Vicia faba* roots. The mitotic index found in this study ranged from 5.4% to 8.8% corresponding to range found in optimal conditions of roots growth and cell proliferation [37]. Our results show a significant genotoxicity in almost half of water samples. Well water, springs and river showed high genotoxic effects compared to the negative control. In line with findings from a study that reported genotoxic activity in ground and surface water for drinking, samples collected from spring and river were the most genotoxic [38, 39]. To our knowledge, there is no study on the water genotoxicity in Cameroon and our study is the first one investigating water genotoxicity using *Vicia faba* seeds in Cameroon including the Central African sub region. The rare genotoxic studies carried-out in Africa were almost done on plants used for traditional medicine using other methods such as comet essay, *Salmonella* microsome assay [40–42]. Our approach has the merit of being simple and reliable. It is especially adapted to study environment quality in areas like West Cameroon where there are intensive agricultural activities. The results have highlighted a potential health risk for the populations who live around this area.

## Conclusion

Water quality monitoring is essential to attest whether water sources are safe for human consumption or not. The use of valid methods for water analysis is also important. Results showed that almost all water samples were found to be contaminated by TC including treated water sources, highlighting the poor quality of domestic water sources in West region of Cameroon and lack of appropriate knowledge regarding water treatment techniques. Some water sources indicated genotoxic effects on *Vicia faba* root tips. Results demonstrated the need to conduct regular monitoring of drinking water sources. Community empowerment on water treatment methods and wastes management need to be implemented to help improve water quality. As previous studies have shown that disinfectants can induce an increase in the water genotoxicity, water treatment should be done following recommended protocols. There is also a need to design adequate faeces disposal systems, mostly around water drinking sources.

## Supporting information

**S1 Dataset.**
(XLS)

## Acknowledgments

The authors express their gratitude to the administrative staff of the Evangelic University of Cameroon.

## Author Contributions

**Conceptualization:** Rodrigue Biguioh Mabvouna,  Sali Ben Béchir Adogaye, Vittorio Colizzi.

**Data curation:** Rodrigue Biguioh Mabvouna,  Sali Ben Béchir Adogaye.

**Formal analysis:** Rodrigue Biguioh Mabvouna.

**Funding acquisition:** Rodrigue Biguioh Mabvouna, Jean Blaise Kemogne, Vittorio Colizzi.

**Investigation:** Rodrigue Biguioh Mabvouna, Andrillene Laure Deutou Wondeu.

**Methodology:** Rodrigue Biguioh Mabvouna,  Sali Ben Béchir Adogaye, Vittorio Colizzi.

**Project administration:** Rodrigue Biguioh Mabvouna.

**Supervision:** Martin Sanou Sobze, Jean Blaise Kemogne, Vittorio Colizzi.

**Validation:** Rodrigue Biguioh Mabvouna,  Sali Ben Béchir Adogaye, Patrick Martial Pete Nkamedjie, Jean Blaise Kemogne, Vittorio Colizzi.

**Writing – original draft:** Rodrigue Biguioh Mabvouna.

**Writing – review & editing:** Rodrigue Biguioh Mabvouna,  Sali Ben Béchir Adogaye, Patrick Martial Pete Nkamedjie, Andrillene Laure Deutou Wondeu, Martin Sanou Sobze, Vittorio Colizzi.

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
