## [Decision Letter · Decision Letter 0]

6 Jul 2020

PONE-D-20-17828

Microbiological quality and genotoxicity of domestic water sources: A combined approach using Micro biological Survey method and mutagenesis assay (micronucleus test) in root tips of Vicia Faba in the West Region of Cameroon

PLOS ONE

Dear Dr. Mabvouna,

Thank you for submitting your manuscript to PLOS ONE. After careful consideration, we feel that it has merit but does not fully meet PLOS ONE’s publication criteria as it currently stands. Therefore, we invite you to submit a revised version of the manuscript that addresses the points raised during the review process.

We look forward to receiving your revised manuscript.

Kind regards,

Swatantra Pratap Singh, Ph.D.

Academic Editor

PLOS ONE

Journal Requirements:

2. In your Methods section, please provide additional details regarding the source of the water samples. Please provide the geographic coordinates of the collection locations to ensure reproducibility of the analyses.

3.We note that [Figure(s) 1] in your submission contain [map/satellite] images which may be copyrighted. All PLOS content is published under the Creative Commons Attribution License (CC BY 4.0), which means that the manuscript, images, and Supporting Information files will be freely available online, and any third party is permitted to access, download, copy, distribute, and use these materials in any way, even commercially, with proper attribution. For these reasons, we cannot publish previously copyrighted maps or satellite images created using proprietary data, such as Google software (Google Maps, Street View, and Earth). For more information, see our copyright guidelines: http://journals.plos.org/plosone/s/licenses-and-copyright.

1.    You may seek permission from the original copyright holder of Figure(s) [1] to publish the content specifically under the CC BY 4.0 license. 

4. Please ensure that you refer to Figure 3 in your text as, if accepted, production will need this reference to link the reader to the figure.

5. Please include a copy of Table 2 and 3 which you refer to in your text on page 17 and 18 (there are two tables uploaded named as Figure 2 and Figure 3).

6. Your ethics statement must appear in the Methods section of your manuscript. If your ethics statement is written in any section besides the Methods, please move it to the Methods section and delete it from any other section. Please also ensure that your ethics statement is included in your manuscript, as the ethics section of your online submission will not be published alongside your manuscript.

7.We note that you have indicated that data from this study are available upon request. PLOS only allows data to be available upon request if there are legal or ethical restrictions on sharing data publicly. For information on unacceptable data access restrictions, please see http://journals.plos.org/plosone/s/data-availability#loc-unacceptable-data-access-restrictions.

Additional Editor Comments (if provided):

The article titled “Microbiological quality and genotoxicity of domestic water sources: A combined approach using Micro biological Survey method and mutagenesis assay (micronucleus test) in root tips of Vicia Faba in the West Region of Cameroon” is interesting, however, it has very high similarity with a previous article on "Microbiological quality of water sources in the West region of Cameroon: quantitative detection of total coliforms using Micro Biological Survey method". This article cannot be accepted in the present form and need Major revision in terms of presentation, methodology, and discussion. The manuscript lacks many technical details that require clarification and/or improvement. Authors are suggested to address the reviews comments in order to make the manuscript suitable for publication and provide point-by-point justification or rebuttal against the comments:

Reviewers' comments:

Reviewer's Responses to Questions

**Comments to the Author**

1. Is the manuscript technically sound, and do the data support the conclusions?

Reviewer #1: No

Reviewer #2: Partly

2. Has the statistical analysis been performed appropriately and rigorously? 

Reviewer #1: No

Reviewer #2: Yes

3. Have the authors made all data underlying the findings in their manuscript fully available?

Reviewer #1: Yes

Reviewer #2: Yes

4. Is the manuscript presented in an intelligible fashion and written in standard English?

Reviewer #1: No

Reviewer #2: No

5. Review Comments to the Author

Reviewer #1: The work has been found to be very interesting, however there are some major suggestions with the respect to the design of the experiment and also discussions in the writeup:

1) Details about the sampling to be given in more depth

2) As the microbial load of the water is to be tested, the time of the day or even the temperature at the time of collection is crucial, please provide the data

3)Why E.coli, please explain the need for ennumeration of the indicator, depth in literature lacking especially in the introduction and discussion

3) Why an enzymatic assay like MBS, why not membrane filtration? Explicitly mention the advantages and the underlying principle and the rationale behind the choice of the method

4)Micronucleus test is a standard method, please cite where it has been adapted from.

5)Flowchart on the protocol for micronucleus test is recommended

6)22 samples may be too few to come up with a valid conclusion

Reviewer #2: Manuscript Number: PONE-D-20-17828

Title: Microbiological quality and genotoxicity of domestic water sources: A combined approach using Micro biological Survey method and mutagenesis assay (micronucleus test) in root tips of Vicia Faba in the West Region of Cameroon

Authors: Rodrigue Biguioh Mabvouna, Sali Ben Béchir Adogaye, Patrick Martial Pete Nkamedjie, Adrillene Laure Wondeu Deutou, Martin Sanou Sobze, Jean Blaise Kemogne and Vittorio Colizzi

Article Type: Research Article

Journal: PLOS ONE

Comments: The manuscript highlights the poor quality of domestic water sources in the Central African sub-region. The study deals with the microbial and genetic toxicity induced factors responsible for deteriorating water quality of different natural/domestic sources of drinking water. While this is worthy of investigation, the manuscript lacks in clarity of presentation, especially the methodology and discussion section. The manuscript provides a good read on detection of total coliforms (TC) which was done using the Micro Biological Survey-HACCP & water Easy test along with micronucleus test to assess the genetic toxicity caused due to the poor water quality. However, the approach of the study lacks many technical details that need clarification and/or improvement. Authors should improve the usage of the English language while revising the manuscript. Authors are suggested to address following comments in order to make the manuscript suitable for publication and provide point-by-point justification or rebuttal against each comment:

1. The content of abstract requires modification. Authors are suggested to make the abstract more clear and informative. Provide concise and factual abstract briefly and sequentially stating the aims (objectives), major results, and principal conclusions of the work. Derive the conclusion at last. Provide the data (values) for significant findings of the present study.

2. Abstract, line 29, and Study design and period, line 123, At each site,…genotoxic analysis. The word ‘litter’ should be replaced by ‘litre’ which is a unit of volume.

3. Introduction, lines 78-80, According to World Health Organization…between life stages. Please rephrase the sentence.

4. Total coliforms detection, Lines 130-131, what does it mean by rapid color change? Specify the changes in color corresponding to a definite range for the determination of increasing or decreasing coliform count.

5. Total coliforms detection, Lines 130-131, which reagent is the author talking about? Please mention.

6. Total coliforms detection, Lines 141-145, please mention the specific color change and concentration range (very high, high and low) for coliform count after 24 hours. Also clarify the reason for the reducing coliform counts with the passage of time.

7. Total coliforms detection, lines 141-145, A color change…103 CFU/ml. Please rephrase the sentence.

8. Of the 4 water sources sampled (as mentioned in study design and discussion), results of drilling water is not mentioned in Table 1. Further, results lack the complete details on TC concentration range along all the sources. Also, provide the coordinates for the sampling locations.

9. The reason for opting/choosing Vicia faba as a study plant is not mentioned in the methodology. Authors need to explain with proper rationale or any background study.

10. MN (micronucleus) test abbreviation must be given in the sub headings rather than providing details within the text body for better understanding of readers. Also brief introduction about MN test is not provided as given for Micro Biological Survey-HACCP & water Easy test for coliforms. Please include.

11. The major criticism of the manuscript lies in the lack of technical details provided for the methodology adopted. Authors are suggested to make this section clearer and explanatory. How well water is supposed to be more contaminated with higher coliform count during rainy reason and not in the surface water source? Explanation provided is not satisfactory and subject to contradiction. More meaningful and concrete justification is required.

12. Discussion, TC concentration detected highest among wells but later genotoxicity effects has been reported in spring water, river water as well as well water. What could be the possible reason for such differences? Authors need to address the same.

13. What specific genotoxicity effects do author referring to?? Whatever genetic changes/ mutation detected in the experiments is nowhere mentioned in the manuscript. Please clarify.

14. Discussion, line 215, “However, the TC concentration of untreated source water was found to be higher than treated water sources with no significant difference” – the sentence itself showing significant differences in TC concentration. Please rephrase suitably.

15. Discussion, lines 217 – 219, (Other standardized……..environmental pollutions) – past studies and future need for more such studies needs to be differentiated well. Rephrase the sentence carefully.

16. Discussion, line 232, intensive agricultural activities instead of “intensives”. Please check the whole manuscript for other grammatical errors.

17. The conclusion section must state the key findings of the study. Authors are suggested to draw major inferences/primary conclusions first from the results followed by the secondary conclusions reached through the critical analysis.

6. PLOS authors have the option to publish the peer review history of their article (what does this mean?). If published, this will include your full peer review and any attached files.

Reviewer #1: No

Reviewer #2: No

---

## [Author Response · Author response to Decision Letter 0]

13 Oct 2020

Responses to general comments 

We are not the first authors of the laboratory protocols used in this study. We have added the references of these protocols in the main manuscript to allow reproducibility of the results. 

A marked-up copy of the manuscript highlighting changes made to the original version was prepared and uploaded separately. An unmarked version of the revised paper without tracked changes was also submitted. 

Our financing statement remains unchanged. 

Academic Editor

Journal Requirements:

Author’s responses: We revised the manuscript according to PLOS ONE style requirements, including file naming.

2. In your Methods section, please provide additional details regarding the source of the water samples. Please provide the geographic coordinates of the collection locations to ensure reproducibility of the analyses.

Author’s responses: A table with sampling location names, water source categories, geographic coordinates and date of sample collection was added in the methods section. 

3. We note that [Figure(s) 1] in your submission contain [map/satellite] images which may be copyrighted. All PLOS content is published under the Creative Commons Attribution License (CC BY 4.0), which means that the manuscript, images, and Supporting Information files will be freely available online, and any third party is permitted to access, download, copy, distribute, and use these materials in any way, even commercially, with proper attribution. For these reasons, we cannot publish previously copyrighted maps or satellite images created using proprietary data, such as Google software (Google Maps, Street View, and Earth). 

Author’s responses: We are the authors of the figure 1. We have added details of the software used to generate the figure in the revised version of the manuscript. 

4. Please ensure that you refer to Figure 3 in your text as, if accepted, production will need this reference to link the reader to the figure.

Author’s responses: The correction has been done. There was confusion between tables and figures.

5. Please include a copy of Table 2 and 3 which you refer to in your text on page 17 and 18 (there are two tables uploaded named as Figure 2 and Figure 3).

Author’s responses: The correction was done. 

6. Your ethics statement must appear in the Methods section of your manuscript. If your ethics statement is written in any section besides the Methods, please move it to the Methods section and delete it from any other section. Please also ensure that your ethics statement is included in your manuscript, as the ethics section of your online submission will not be published alongside your manuscript.

Author’s responses: The ethics statement was added in the Methods section and included in the manuscript. 

7. We note that you have indicated that data from this study are available upon request. PLOS only allows data to be available upon request if there are legal or ethical restrictions on sharing data publicly. For information on unacceptable data access restrictions, please see http://journals.plos.org/plosone/s/data-availability#loc-unacceptable-data-access-restrictions.

Author’s responses: We change our declaration; the data sets used and/or analysed during the current study are available without restrictions. 

Author’s responses: There are no ethical or legal restrictions on sharing data set. Data can be viewed without restrictions.

b) If there are no restrictions, please upload the minimal anonymized data set necessary to replicate your study findings as either Supporting Information files or to a stable, public repository and provide us with the relevant URLs, DOIs, or accession numbers. 

Author’s responses: We changed our Data Availability statement and we will upload the study data set as Supporting Information files. 

Additional Editor Comments

The article titled “Microbiological quality and genotoxicity of domestic water sources: A combined approach using Micro biological Survey method and mutagenesis assay (micronucleus test) in root tips of Vicia Faba in the West Region of Cameroon” is interesting, however, it has very high similarity with a previous article on "Microbiological quality of water sources in the West region of Cameroon: quantitative detection of total coliforms using Micro Biological Survey method". This article cannot be accepted in the present form and need Major revision in terms of presentation, methodology, and discussion. The manuscript lacks many technical details that require clarification and/or improvement. Authors are suggested to address the reviews comments in order to make the manuscript suitable for publication and provide point-by-point justification or rebuttal against the comments:

Author’s responses: Here, we provide a point-by-point response to the comments made by reviewers. 

In the first study "Microbiological quality of water sources in the West region of Cameroon: quantitative detection of total coliforms using Micro Biological Survey method" we applied the MBS method as simple and rapid test to check water potability. In this second study, we combined the MBS and micronucleus test using Vicia faba seeds to propose an approach for bacteriological and chemical routine quality check of drinking water which can be easily implemented in developing countries. 

Comments to the Author

1. Is the manuscript technically sounds, and do the data support the conclusions?

Reviewer #1: No

Reviewer #2: Partly

Author’s responses: All experiments conducted in this study were done according the standard's protocols (citations added in the main manuscript). We have further commented on our results to support our conclusion. 

2. Has the statistical analysis been performed appropriately and rigorously?

Reviewer #1: No

Reviewer #2: Yes

Author’s responses: Statistical analyses were done using Stata IC/15.0 software (College Station, Texas 77845 USA, http://www.stata.com), and the main results were presented and interpreted. 

3. Have the authors made all data underlying the findings in their manuscript fully available?

The PLOS Data policy requires authors to make all data underlying the findings described in their manuscript fully available without restriction, with rare exception (please refer to the Data Availability Statement in the manuscript PDF file). The data should be provided as part of the manuscript or its supporting information or deposited to a public repository. For example, in addition to summary statistics, the data points behind means, medians and variance measures should be available. If there are restrictions on publicly sharing data—e.g. participant privacy or use of data from a third party—those must be specified.

Reviewer #1: Yes

Reviewer #2: Yes

Authors responses: Nothing to add

4. Is the manuscript presented in an intelligible fashion and written in standard English?

Reviewer #1: No

Reviewer #2: No

Author’s responses: The manuscript was submitted to an English-speaking person for language revision and improves its quality. 

5. Review Comments to the Author

Please use the space provided to explain your answers to the questions above. You may also include additional comments for the author, including concerns about dual publication, research ethics, or publication ethics. (Please upload your review as an attachment if it exceeds 20,000 characters). 

Authors responses: 

Reviewer reports 

Reviewer #1

The work has been found to be very interesting; however, there are some major suggestions with the respect to the design of the experiment and also discussions in the write-up:

1) Details about the sampling to be given in more depth

Author’s responses: More details in the sampling procedure of water sources were added in the manuscript. Water was collected (in the morning before 12 o’clock), we didn’t apply a statistical sampling method to identify water source to be collected. The sources of water were identified step by step according to their availability, accessibility, and their use as drinking water by the local populations.

2) As the microbial load of the water is to be tested, the time of the day or even the temperature at the time of collection is crucial, please provide the data

Author’s responses: The date and time of water sampling were provided in the manuscript. Temperature and pH of water were measured before leaved the site. The temperatures and pH didn’t significatively vary between water sources. The averages values of both measurements were given in the results section.

3) Why E. coli, please explain the need for enumeration of the indicator, depth in literature lacking especially in the introduction and discussion

Author’s responses: We didn’t enumerate E. coli but Total Coliforms bacteria which include Escherichia coli (major species in the faecal coliform group). The MBS test doesn’t permit to clearly identify the bacteria present in sample water. We provided the justification of using TC as water pollution indicator in the manuscript. 

4) Why an enzymatic assay like MBS, why not membrane filtration? Explicitly mention the advantages and the underlying principle and the rationale behind the choice of the method

Author’s responses: We mentioned in the main manuscript the advantages of MBS and cited previous studies which applied this test for water analysis in Africa and in Cameroon. 

5) Micronucleus test is a standard method, please cite where it has been adapted from

Author’s responses: We provided the citation in the text.

6) Flowchart on the protocol for micronucleus test is recommended

Author’s responses: We described all the steps of the MN analysis in the Methods section and we cited the source of the protocol. We think that these are sufficient for the results reproducibility. 

7) 22 samples may be too few to come up with a valid conclusion

Author’s responses: As we mentioned above, the sources of water to be sampled were identified step by step according to their availability, accessibility. It is also important to note that we are in context of water scarcity. 

Reviewer #2

Comments: The manuscript highlights the poor quality of domestic water sources in the Central African sub-region. The study deals with the microbial and genetic toxicity induced factors responsible for deteriorating water quality of different natural/domestic sources of drinking water. While this is worthy of investigation, the manuscript lacks in clarity of presentation, especially the methodology and discussion section. The manuscript provides a good read on detection of total coliforms (TC) which was done using the Micro Biological Survey-HACCP & water Easy test along with micronucleus test to assess the genetic toxicity caused due to the poor water quality. However, the approach of the study lacks many technical details that need clarification and/or improvement. Authors should improve the usage of the English language while revising the manuscript. Authors are suggested to address following comments in order to make the manuscript suitable for publication and provide point-by-point justification or rebuttal against each comment. 

Author’s responses: We did our best to address the comments and improve the quality of the manuscript. 

1. The content of abstract requires modification. Authors are suggested to make the abstract more clear and informative. Provide concise and factual abstract briefly and sequentially stating the aims (objectives), major results, and principal conclusions of the work. Derive the conclusion at last. Provide the data (values) for significant findings of the present study

Author’s responses: The abstract was revised as recommended. 

2. Abstract, line 29, and Study design and period, line 123, At each site,…genotoxic analysis. The word ‘litter’ should be replaced by ‘litre’ which is a unit of volume

Author’s responses: Done 

3. Introduction, lines 78-80, According to World Health Organization…between life stages. Please rephrase the sentence

Author’s responses: Done 

4. Total coliforms detection, Lines 130-131, what does it mean by rapid color change? Specify the changes in color corresponding to a definite range for the determination of increasing or decreasing coliform count. 

Author’s responses: A rapid color change (from red to yellow) indicates a very high concentration of coliforms in the water sample. We rephrased the sentence. The change in color according to the time of analyses and the corresponding concentrations of coliforms are given in the main manuscript. 

5. Total coliforms detection, Lines 130-131, which reagent is the author talking about? Please mention

Author’s responses: We are talking about test, not a reagent. The MBS kit comes in a pack containing all the material necessary for the analysis without need additional reagent. More details on MBS-HACCP & water Easy test are given in the main manuscript. 

6. Total coliforms detection, Lines 141-145, please mention the specific color change and concentration range (very high, high, and low) for coliform count after 24 hours. Also clarify the reason for the reducing coliform counts with the passage of time.

Author’s responses: There is one change in color from red to yellow (contamination). The results were interpreted according to the standard protocol. We specified the level of Coliforms concentration according to the time required for the color change. 12h = very high contamination; 19h = high contamination and 24h = low contamination. The color change is proportional to the concentration of coliforms in the water samples, the greater the number of microorganisms, the more rapid the change of color, thus, a positive result (contamination).

7. Total coliforms detection, lines 141-145, A color change…103 CFU/ml. Please rephrase the sentence.

Author’s responses: Done 

8. Of the 4 water sources sampled (as mentioned in study design and discussion), results of drilling water is not mentioned in Table 1. Further, results lack the complete details on TC concentration range along all the sources. Also, provide the coordinates for the sampling locations.

Author’s responses: Results of drilling water were mentioned; we used a synonym, now we cleared mentioned drilling water in all tables. We provide all result details available when using MBS method. The most important result is the indication of whether TC bacteria are present or not and the level of concentration. The coordinates of sampling locations were provided in the manuscript. 

9. The reason for opting/choosing Vicia faba as a study plant is not mentioned in the methodology. Authors need to explain with proper rationale or any background study.

Author’s responses: The reasons for opting/choosing Vicia faba were given in the Methods section. 

10. MN (micronucleus) test abbreviation must be given in the sub headings rather than providing details within the text body for better understanding of readers. Also brief introduction about MN test is not provided as given for Micro Biological Survey-HACCP & water Easy test for coliforms. Please include.

Author’s responses: Done 

11. The major criticism of the manuscript lies in the lack of technical details provided for the methodology adopted. Authors are suggested to make this section clearer and explanatory. How well water is supposed to be more contaminated with higher coliform count during rainy reason and not in the surface water source? Explanation provided is not satisfactory and subject to contradiction. More meaningful and concrete justification is required

Author’s responses: Additional details on the methods used were given, and we specified (and cited) that we used the standard protocols. The analyses used in this study are simple to execute and don’t require sophisticated instruments or additional reagent (MBS method). We provide all the steps of our methodology necessary for the reproducibility of the results. 

We are talking about concentration (quantity) of bacteria in well water. A previous study conducted in the same area (Bafoussam), during the same season found similar results. We think that bacteria growth is most high in groundwater than surface water (running water), and the combined effect of temperatures (from 22 to 26 °C) and pH (5.5 to 8.3) could favour bacteria growth. The proximity between wells and latrines could also favour contamination (by runoff water or slick phreatic communication). We added other justifications in the manuscript. 

12. Discussion, TC concentration detected highest among wells, but later genotoxicity effects has been reported in spring water, river water as well as well water. What could be the possible reason for such differences? Authors need to address the same.

Author’s responses: There is no relation between genotoxicity and TC concentration, and results are coherent. If genotoxic effect is detected in water this indicates the presence of chemicals (disinfectant, heavy metal, etc.), and generally there are toxic for bacteria or limit bacteria growth (low concentration). Finally we noted in the study that some wells were chlorinated (that’s why 1 ml of Thiosulphate was added to water sample to neutralize the chlorine), so it’s consistent if genotoxic effects are also detected in well water. 

13. What specific genotoxicity effects do author referring to?? Whatever genetic changes/ mutation detected in the experiments is nowhere mentioned in the manuscript. Please clarify.

Author’s responses: Micronucleus as mutagenesis endpoint (chromosome fragments). We clarified in the manuscript. 

14. Discussion, line 215, “However, the TC concentration of untreated source water was found to be higher than treated water sources with no significant difference” – the sentence itself showing significant differences in TC concentration. Please rephrase suitably.

Author’s responses: No statistically significant difference at the significance level set. We rephrased. 

15. Discussion, lines 217 – 219, (Other standardized……..environmental pollutions) – past studies and future need for more such studies needs to be differentiated well. Rephrase the sentence carefully.

Author’s responses: Done 

16. Discussion, line 232, intensive agricultural activities instead of “intensives”. Please check the whole manuscript for other grammatical errors

Author’s responses: Done 

17. The conclusion section must state the key findings of the study. Authors are suggested to draw major inferences/primary conclusions first from the results followed by the secondary conclusions reached through the critical analysis.

Authors responses: Done

---

## [Decision Letter · Decision Letter 1]

25 Nov 2020

PONE-D-20-17828R1

Microbiological quality and genotoxicity of domestic water sources: A combined approach using Micro biological Survey method and mutagenesis assay (micronucleus test) in root tips of Vicia Faba in the West Region of Cameroon

PLOS ONE

Dear Dr. Mabvouna,

Thank you for submitting your manuscript to PLOS ONE. After careful consideration, we feel that it has merit but does not fully meet PLOS ONE’s publication criteria as it currently stands. Therefore, we invite you to submit a revised version of the manuscript that addresses the points raised during the review process.

We look forward to receiving your revised manuscript.

Kind regards,

Swatantra Pratap Singh, Ph.D.

Academic Editor

PLOS ONE

Additional Editor Comments (if provided):

The article has been improved, however few miner corrections are required before publication as suggested by the reviewers.

Reviewers' comments:

Reviewer's Responses to Questions

**Comments to the Author**

1. If the authors have adequately addressed your comments raised in a previous round of review and you feel that this manuscript is now acceptable for publication, you may indicate that here to bypass the “Comments to the Author” section, enter your conflict of interest statement in the “Confidential to Editor” section, and submit your "Accept" recommendation.

Reviewer #1: All comments have been addressed

Reviewer #2: All comments have been addressed

2. Is the manuscript technically sound, and do the data support the conclusions?

Reviewer #1: Partly

Reviewer #2: Yes

3. Has the statistical analysis been performed appropriately and rigorously? 

Reviewer #1: Yes

Reviewer #2: Yes

4. Have the authors made all data underlying the findings in their manuscript fully available?

Reviewer #1: No

Reviewer #2: Yes

5. Is the manuscript presented in an intelligible fashion and written in standard English?

Reviewer #1: Yes

Reviewer #2: Yes

6. Review Comments to the Author

Reviewer #1: The work done is interesting however some suggestions on improving the clarity of the writeup is recommended:

Line 130: 1 mL of what concentration of thiosulphate? Is it sodium thiosulphate, if yes please specify, if not please mention the salt name explicitly

Line 133: Mention explicitly the temperature and pH were taken before or after placing the samples in the ice box

Table 1: The temperature mentioned is for the water or ambient air temperature? Also, mentioning the time of collection is suggested

Line 160-164: Add the reference

For the section of total coliform detection: The test details insufficient, which enzymes play a crucial role in the color change? More details on the mechanism of color change needs to be explained to understand how the kit works?

Line 219: Why so much variation in the pH, was it because the samples were collected at different time frames of the day, was the time of collection not kept constant? Or was there appearance or disappearance of algal blooms?

Reviewer #2: The manuscript has been revised in accordance with the comments provided. Authors have also satisfactorily answered and clarified the doubts raised. The revised manuscript is now suitable for acceptance and publication in the Journal.

7. PLOS authors have the option to publish the peer review history of their article (what does this mean?). If published, this will include your full peer review and any attached files.

Reviewer #1: No

Reviewer #2: No

---

## [Author Response · Author response to Decision Letter 1]

4 Dec 2020

Additional Editor Comments (if provided): 

The article has been improved; however few miner corrections are required before publication as suggested by the reviewers.

Author’s responses: Corrections as suggested by the reviewers have been done (see below).

Reviewers' comments: 

Reviewer's Responses to Questions

Comments to the Author: 

1. If the authors have adequately addressed your comments raised in a previous round of review and you feel that this manuscript is now acceptable for publication, you may indicate that here to bypass the “Comments to the Author” section, enter your conflict of interest statement in the “Confidential to Editor” section, and submit your "Accept" recommendation.

Reviewer #1: All comments have been addressed

Reviewer #2: All comments have been addressed

Author’s responses: nothing to report

2. Is the manuscript technically sound, and do the data support the conclusions?

Reviewer #1: Partly

Reviewer #2: Yes

Author’s responses: We added more information in the revised manuscript as requested by Reviewer #1 

3. Has the statistical analysis been performed appropriately and rigorously? 

Reviewer #1: Yes

Reviewer #2: Yes

Author’s responses: nothing to report

4. Have the authors made all data underlying the findings in their manuscript fully available?

Reviewer #1: No

Reviewer #2: Yes

Author’s responses: We already change our data availability statement: Data used for this study are fully available without restriction and the data set was uploaded as supporting information during the revision process. 

5. Is the manuscript presented in an intelligible fashion and written in standard English?

Reviewer #1: Yes

Reviewer #2: Yes

Author’s responses: nothing to report

6. Review Comments to the Author

Author’s comments: 

It’s not a dual publication. As we already responded, in the study "Microbiological quality of water sources in the West region of Cameroon: quantitative detection of total coliforms using Micro Biological Survey method" we applied the MBS method as simple and rapid test to check water potability. In this second study, we combined the MBS and micronucleus test using Vicia faba seeds to propose an approach for bacteriological and chemical routine quality check of drinking water which can be easily implemented in developing countries.

Regarding the research ethics, or publication ethics, our manuscript does not report on or involve the use of any animal or human data or tissue, so an ethics approval and consent are not required with reference to Order No. 079/A/MSP/DS of the Minister of Public Health of October 22, 1987 establishing and organizing an Ethics Committee on Research Involving Human Beings (article 2).

Reviewer #1: 

The work done is interesting however some suggestions on improving the clarity of the writeup is recommended:

1. Line 130: 1 mL of what concentration of thiosulphate? Is it sodium thiosulphate, if yes please specify, if not please mention the salt name explicitly

Author’s responses: It is 1 ml of Sodium Thiosulfate solution 10% (0.25g/50ml). We specified in the revised manuscript. 

2. Line 133: Mention explicitly the temperature and pH were taken before or after placing the samples in the ice box

Author’s responses: Done: Temperature and pH of water were measured on site before placing the samples in the ice box. We mentioned explicitly it the revised manuscript. 

3. Table 1: The temperature mentioned is for the water or ambient air temperature? Also, mentioning the time of collection is suggested

Author’s responses: The temperature mentioned is for the water samples, we specified it in the revised manuscript. We did not accurately record the time of sample collection, but as mentioned in the revised manuscript, sample collection was done in the morning before 12 o’clock.

4. Line 160-164: Add the reference

Author’s responses: Done!

5. For the section of total coliform detection: The test details insufficient, which enzymes play a crucial role in the color change? More details on the mechanism of color change needs to be explained to understand how the kit works?

Author’s responses: MBS method is a colorimetric reaction system based on the measurement of the catalytic activity of oxidoreductase enzymes of primary metabolism, which allows you to determine a correspondence between microorganism and enzyme activity in the sample. This precision was added in the revised manuscript. 

6. Line 219: Why so much variation in the pH, was it because the samples were collected at different time frames of the day, was the time of collection not kept constant? Or was there appearance or disappearance of algal blooms?

Author’s responses: Some factors can explain the large variation of the water pH, such as time of samples collection. But think that we have minimized this influence since all the samples were collected in the same period (in the morning before 12 o'clock). Rather, we think that this variation of the pH may be due to the difference in depth of the water sources (wells, water drilling) or to the composition of the underground rock. We will take these factors into account in future studies.

Reviewer #2: The manuscript has been revised in accordance with the comments provided. Authors have also satisfactorily answered and clarified the doubts raised. The revised manuscript is now suitable for acceptance and publication in the Journal.

Author’s responses: nothing to report

7. PLOS authors have the option to publish the peer review history of their article (what does this mean?). If published, this will include your full peer review and any attached files.

Do you want your identity to be public for this peer review? For information about this choice, including consent withdrawal, please see our Privacy Policy.

Reviewer #1: No

Reviewer #2: No

Author’s responses: nothing to report

---

## [Decision Letter · Decision Letter 2]

30 Dec 2020

Microbiological quality and genotoxicity of domestic water sources: A combined approach using Micro biological Survey method and mutagenesis assay (micronucleus test) in root tips of Vicia Faba in the West Region of Cameroon

PONE-D-20-17828R2

Dear Dr. Mabvouna,

We’re pleased to inform you that your manuscript has been judged scientifically suitable for publication and will be formally accepted for publication once it meets all outstanding technical requirements.

Kind regards,

Swatantra Pratap Singh, Ph.D.

Academic Editor

PLOS ONE

Additional Editor Comments (optional):

As per reviewers comments and we please to accept the article for the publication.

Reviewers' comments:

Reviewer's Responses to Questions

**Comments to the Author**

1. If the authors have adequately addressed your comments raised in a previous round of review and you feel that this manuscript is now acceptable for publication, you may indicate that here to bypass the “Comments to the Author” section, enter your conflict of interest statement in the “Confidential to Editor” section, and submit your "Accept" recommendation.

Reviewer #1: All comments have been addressed

Reviewer #2: All comments have been addressed

2. Is the manuscript technically sound, and do the data support the conclusions?

Reviewer #1: Yes

Reviewer #2: Yes

3. Has the statistical analysis been performed appropriately and rigorously? 

Reviewer #1: Yes

Reviewer #2: Yes

4. Have the authors made all data underlying the findings in their manuscript fully available?

Reviewer #1: Yes

Reviewer #2: Yes

5. Is the manuscript presented in an intelligible fashion and written in standard English?

Reviewer #1: Yes

Reviewer #2: Yes

6. Review Comments to the Author

Reviewer #1: The paper has covered a very important topic and the authors have tried and included all the revisions suggested by the reviewers

Reviewer #2: The manuscript has been revised suitably. The revised manuscript is now suitable for acceptance and publication in the Journal.

7. PLOS authors have the option to publish the peer review history of their article (what does this mean?). If published, this will include your full peer review and any attached files.

Reviewer #1: No

Reviewer #2: No

---

## [Editor Report · Acceptance letter]

12 Jan 2021

PONE-D-20-17828R2 

Microbiological quality and genotoxicity of domestic water sources: A combined approach using Micro biological Survey method and mutagenesis assay (micronucleus test) in root tips of *Vicia faba* in the West Region of Cameroon 

Dear Dr. Mabvouna:

I'm pleased to inform you that your manuscript has been deemed suitable for publication in PLOS ONE. Congratulations! Your manuscript is now with our production department. 

Kind regards, 

on behalf of

Dr. Swatantra Pratap Singh 

Academic Editor

PLOS ONE